# A thorough Investigation of Rare-Earth Dy^3+^ Substituted Cobalt-Chromium Ferrite and Its Magnetoelectric Nanocomposite

**DOI:** 10.3390/nano13071165

**Published:** 2023-03-24

**Authors:** Ram H. Kadam, Ravi Shitole, Santosh B. Kadam, Kirti Desai, Atul P. Birajdar, Vinod K. Barote, Khalid Mujasam Batoo, Sajjad Hussain, Sagar E. Shirsath

**Affiliations:** 1Materials Research Laboratory, Srikrishna Mahavidyalaya Gunjoti, Omerga 413613, India; 2Department of Physics, Lal Bahadur Shastri Senior College, Partur 431501, India; 3Department of Physics, Balbhim College, Beed 431122, India; 4Department of Physics, B.S.S. Arts, Science and Commerce College, Makni 413606, India; 5Department of Physics, Sant Dnyaneshwar Mahavidyalaya, Soegaon 431120, India; 6College of Science, King Saud University, P.O. Box 2455, Riyadh 11451, Saudi Arabia; 7Graphene Research Institute and Institute of Nano and Advanced Materials Engineering, Sejong University, Seoul 143-747, Republic of Korea; 8School of Materials Science and Engineering, University of New South Wales, Sydney, NSW 2052, Australia

**Keywords:** ferrites, Rietveld refinement, strain mechanism, elastic properties, magnetic properties, magnetoelectric properties

## Abstract

The stoichiometric compositions of a ferrite system with a chemical formula CoCr_0.5_Dy_x_Fe_1.5−x_O_4_ where x = 0.0, 0.025, 0.05, 0.075 and 0.1 were prepared by the sol-gel auto-combustion method. The structural, morphological and magnetic properties were studied by the X-ray diffraction (XRD), infra-red spectroscopy (IR), scanning electron microscopy, transmission electron microscopy and vibrating sample magnetometer. XRD analysis confirmed the cubic spinel structure of the prepared samples without the presence of any impurity and secondary phases. Selected area electron diffraction and IR measurements gives further confirmation to the XRD observations. Considering that strain mechanism, elastic properties and cation distribution play a major role for controlling the magnetic properties and therefore these properties were precisely evaluated through reliable methodologies such as XRD and IR data. The cation distribution was determined by the X-ray diffraction data which are further supported by the magnetization studies. Magnetoelectric properties of CoCr_0.5_Dy_x_Fe_1.5−x_O_4_ + BaTiO_3_ have also been investigated. The mechanisms involved are discussed in the manuscript.

## 1. Introduction

Functionalized nanomaterials gained a lot of attention in the recent years because of their applications in various technological fields [1,2]. Magnetoelectric (ME) materials having coupling between magnetic and electric characteristics have attracted considerable interest from the emergence of large ME coefficient in many composite materials. ME materials have the potential to be applicable in multifunctional devices [3,4,5,6]. Mixed phase ME composite have the proper combination of ferro/ferrimagnetic (magnetostrictive) with ferroelectric (electrostrictive) material which is responsible to generate ME effect. A ferrimagnetic inverse spinel cobalt ferrite (CoFe_2_O_4_:CFO) as a magnetostrictive component in ME materials is one of the significant members of the ferrite class from a commercial point of view [7]. CFO has a high saturation magnetization (400 emu/cm^3^/80 emu/g) and a magnetic easy axis along the [100] directions [8]. Additionally, it has a positive first-order magnetocrystalline anisotropy constant (*K*_1_~2 × 10^6^ ergs/cm^3^) which is orders of magnitude higher than that of other ferrites with a spinel structure leading to a high coercivity (*Hc*). Due to these characteristics, CFO stands out as a special ferrimagnetic-magnetostrictive material in ME composite for magnetic spin filters and spintronics components such as charge-/strain-driven multiferroic nanostructures [9,10,11,12].

CoFe_2_O_4_ with chromium (Cr^3+^) substitution, where Cr ions were used in place of Fe ions possess higher coercivity. However, the lower magnetic moment of Cr that replaced Fe ions at the octahedral B-site caused the saturation magnetization to decrease [13]. Because the rare earth elements have a strong magnetic moment, they can be substituted to prevent this loss of magnetization [14]. Additionally, it has been discovered that minor replacements of Fe ions with rare earth elements may favourably affect ferrite’s magnetic and electrical properties, making it feasible to create a good magnetic material for its use in high frequency applications. The characteristics of ferrite were altered by the addition of trace amounts of rare earth (RE^3+^) ions such as La, Ho, Dy, Gd, Tb, Nd, Sm, Er, Ce and Yb [15,16]. Wang Jing et al. [17] reached the conclusion that Dy^3+^ incorporated W-type ferrite compound has superior microwave absorption characteristics. Additionally, because rare earth ions typically exhibit stronger spin-orbital coupling (SOC) than first row transition metal ions [18], the partial replacement of Fe^3+^ by RE^3+^-Dy^3+^ ions in CoFe_2_O_4_ nanoparticles would result in higher magnetization and coercivity. This is especially true when RE^3+^-Dy^3+^ are anisotropic.

Materials with a high magnetic ordering temperature, a high magnetization and a strong magnetocrystalline anisotropy may be induced from the mixture of the three classes of compounds (Co, Cr), (R_2_O_3_) and (Fe) as a single compound. These compounds with the carefully developed formula; CoCr_0.5_Dy_x_Fe_1.5−x_O_4_, are expected to have a wide range of applicabilities, particularly in the field of advanced technology. The effectiveness of the electronic components constructed of such form of ferrite material is supposed to sustain and work efficiently for a longer period of time. Herein, we are reporting a thorough investigation of the structural, elastic and magnetic properties of CoCr_0.5_Dy_x_Fe_1.5−x_O_4_. Furthermore, ME property of specific compounds of CoCr_0.5_Dy_x_Fe_1.5−x_O_4_ + BaTiO_3_ was investigated to obtain the ME coefficient. We have chosen three compositions of CoCr_0.5_Dy_x_Fe_1.5−x_O_4_ (x = 0.0, 0.05 and 0.1) as a ferromagnetic-magnetostrictive components among the studied samples depending upon their different magnitude of magnetization and coercivity. BaTiO_3_ was selected as ferroelectric materials because of its well-known ferroelectric properties.

In general, controllable and rational processing determine the morphology of nanoparticles, therefore attention should have been given to improve the controllability and designability of nanoparticle preparation [19,20]. Therefore, in the present work, the samples were synthesized by the sol-gel auto-combustion method considering that sol-gel method has numerous advantages, such as low temperature, cost-effective, requiring less time and fast reaction time [21,22,23].

## 2. Materials and Methods

The nano-crystalline samples of the series CoCr_0.5_Dy_x_Fe_1.5−x_O_4_ where x = 0.0, 0.025, 0.05, 0.075 and 0.1 were synthesized by the sol-gel auto-combustion method. A.R. grade citric acid (C_6_H_8_O_7_·H_2_O), cobalt nitrate (Co(NO_3_)_2_⋅6H_2_O), dysprosium(III) nitrate Dy(NO_3_)_3_·xH_2_O chromium nitrate (Cr(NO_3_)_3_⋅9H_2_O) and ferric nitrate (Fe(NO_3_)_3_⋅9H_2_O) (>99% sd-fine) were used as starting materials. The products of the system were produced by keeping metal nitrate to citrate ratio 1:3. Reaction procedure was carried out in air atmosphere without protection of inert gases. The metal nitrates were dissolved together in a minimum amount of double distilled water to obtain a clear solution. An aqueous solution of citric acid was mixed with metal nitrates solution. The mixed solution was kept on to a hot plate with continuous stirring at 100 °C. During evaporation, the solution became viscous and finally formed a very viscous brown gel. When finally all water molecules were removed from the mixture, the viscous gel began frothing. After few minutes, the gel automatically ignited and burnt with glowing flints. The decomposition reaction would not stop before the whole citrate complex was consumed. The auto-ignition was completed within a minute, yielding the brown colour ashes termed as a precursor. The as-prepared powders of all the samples were heat treated separately at 800 °C for 4 h to obtain the final product.

The samples were X-ray examined by Phillips X-ray diffractometer (Model 3710) using Cu K_α_ radiation (λ = 1.5405 Å). The scanning rate was 1.5 °/min and scanning step was 0.02°. The microstructure and morphology of sintered powder were characterized by scanning electron microscopy (SEM) on a JEOL-JSM-5600 N scanning electron microscope (SEM). Stoichiometric proportion of the elements in the composition was analysed using energy dispersive X-ray analysis (EDAX). High resolution transmission electron microscopy (HRTEM, Model CM 200, Philips make) was used to investigate the nanostructure analysis of the prepared samples. The infrared spectra of all the samples were recorded at room temperature in the range 300–800 cm^−1^ on a Perkin Elemer spectrometer (model 783). To study the IR spectra of all the samples, about one gram of fine powder of each sample was mixed with KBr in the ratio 1:250 by weight to ensure uniform distribution in the KBr pellet. The mixed powder was then pressed in a cylindrical die to obtain the circular disc of approximately 1 mm thickness.

Magnetic measurements were performed using the commercial PARC EG&G vibrating sample magnetometer VSM 4500. Magnetic hysteresis loops were measured at room temperature with maximum applied magnetic fields up to 15 kOe.

The magnetoelectric measurements were carried out at room temperature with simultaneous application of constant and alternating magnetic fields on the samples, which were previously electrically polarized in the mode of 1.5 kv/cm at 200 °C for 1 h. The output ME voltage of the sample was measured as a function of bias magnetic field (*H_DC_*) from 0 Oe to 8000 Oe in steps of 500 Oe with sinusoidal magnetic field (*H_AC_* = 5 Oe, *f* = 1 kHz). The ME coefficient was calculated by using the relation:(1)$$\alpha_{ME}=\frac{V}{H_{AC}d}$$
where *V* is voltage generated due to ME effect, *H_AC_* is amplitude of sinusoidal magnetic field and *d* is thickness.

## 3. Results

### 3.1. Structural Aspects

Figure 1 display the Rietveld refined X-ray diffraction patterns (XRD) of CoCr_0.5_Dy_x_Fe_1.5−x_O_4_. The examination of Rietveld refined XRD patterns is associated to the reliability factors viz. profile factor (*R_P_*), R expected (*R_Exp_*), weighted profile factor (*R_WP_*) and goodness factor (*χ*^2^) which are given in Table 1. XRD peaks related to certain crystallographic planes were indexed using the Bragg’s law. These indexed planes correspond to the cubic spinel structure. No secondary and impurity phases were detected in any of the studied samples due to the complete incorporation of rare-earth Dy^3+^ ions in CoCr_0.5_Fe_1.5_O_4_ cubic spinel matrix. The lattice constant ‘*a*’ was obtained for all the observed crystallographic planes and average of them are given in Table 1. The Nelson–Riley (NR) extrapolation function for each lattice reflection of each sample was carried out to obtain the true value of lattice constant “*a*_0_” [24]:(2)$$F\left(\theta \right)=\frac{1}{2}\left[\left(\frac{{cos}^{2}\theta }{sin\theta }\right)+\left(\frac{{cos}^{2}\theta }{\theta }\right)\right]$$

Lattice constant “*a*” for each value of *x* are plotted as a function of *F*(*θ*) (Figure 2). By projecting *F*(*θ*) = 0 or *θ* = 90°, it is simple to determine the true lattice parameter “*a*_0_”. The true value of the “*a*_0_”, differs slightly from the average value, “*a*”, as seen in Table 1 and Table 2. Both “*a*_0_” and “*a*” have shown increment with the Dy^3+^ substitution which is an indication that Dy^3+^ entered into the Co-Cr spinel ferrite lattice and occupied the crystallographic sites.

The Scherrer method, a representation of the Bragg’s line broadening [25], was used to determine the average crystallite size (*t_xrd_*) (Table 2). The Williamson–Hall method is another method that provides superior analysis of crystallite size and the contributions of microstrains associated with the crystal lattice [26]. XRD peak broadening and crystallite size are correlated with each other since peak broadening is caused by the lattice strain alone, or by the combined effects of lattice strain and crystallite/particle size [27,28]. The following equations can be used to correct the observed and instrumental broadening [29,30]:(3)$${\beta^{2}}_{Obs.}={\beta^{2}}_{Size}+{\beta^{2}}_{Strain}+{\beta^{2}}_{Inst.}$$
(4)$${\beta^{2}}_{Tot.}={\beta^{2}}_{Obs.}+{\beta^{2}}_{Inst.}$$
(5)$$\mathrm{Or}\;{\beta^{2}}_{Tot.}={\beta^{2}}_{Strain}+{\beta^{2}}_{Size}$$

Using the values of *β_Strain_* and *β_Size_*; Equation (4) is modified as [31]:(6)$$\beta cos\theta =\left(\frac{k\lambda }{t_{W\text{-}H}}\right)+4\varepsilon sin\theta$$
where *t_W-H_* is the crystallite size determined by the W-H analysis, *k* is the shape factor and *ε*- is the microstrain introduced into the crystal lattice. The W-H equation (Equation (6)) becomes the Debye–Scherrer equation for the zero-strain (unstrain) values [32]. βcosθ vs. 4εsinθ curves (Figure 3) were used to determine the crystallite size (intercept = *k/t_W-H_*) and crystal lattice strain (*ε* = slope) [32]. The crystallite size obtained from the W-H approach varies in the range of 14.9−13.4 nm, whereas it is obtained from the Scherrer relation ranging from14.5 to 12.8 nm.

The substitution of Dy^3+^ ions reduces the *t_W-H_*, which is in line with the findings of the Scherrer method (Table 2). According to the slope of W-H plots, Co-Cr ferrite exhibits positive nature of the microstrains in the crystal lattice. Such positive nature of microstrain refer to the tensile type of strain [33,34]. As seen from Figure 3 and Table 2, the lattice strain (tensile type) was increased from 1.54 × 10^−4^ to 2.96 × 10^−4^. The increase in the lattice constant is mostly responsible for the increase in the tensile strain in Co-Cr ferrite with Dy substitution. The ionic radius of Dy^3+^ ions (0.99Å) is larger compared to Fe^3+^ ions (0.67Å) and thus brought expansion in crystallographic structure of Co-Cr ferrite.

According to Williamson–Hall analysis, the isotropic nature of peak broadening is the key to obtain the lattice strain and crystallite size. This suggests that the diffracting domains are isotropic because of the contribution of the microstrain. Another technique called as size-strain plots (SSP) that can provide more accurate information about the strain-size parameters. The fundamental benefit of this strategy is that less weightage is given to high-angle reflections, whose accuracy level is significantly lower. It is assumed that XRD reflections are overlapped at higher angle, and thus SSP approach predicated that the size profile can be estimated using a Lorentzian function, and the strain profile can be determined using the following Gaussian function relation [35]:(7)$${\left(d_{hkl}\beta_{hkl}cos\theta \right)}^{2}=\frac{k\lambda }{t}\left(d_{hkl}^{2}\beta_{hkl}cos\theta \right)+{\left(\frac{\varepsilon }{2}\right)}^{2}$$

Plots are drawn between $\left(d_{hkl}^{2}\beta_{hkl}cos\theta \right)$ and ${\left(d_{hkl}\beta_{hkl}cos\theta \right)}^{2}$ are shown in Figure 4. Reciprocal of the slopes gives the crystallite size ‘*t_ssp_*’ and the root mean square of Y-intercept gives the strain (Table 2).

The experimental density (*ρ_exp_*) was calculated using the Archimedes’ principle [36]: *ρ_exp_* = weight of the sample in air/loss of weight in the xylene. The formula *ρ_x-ray_* = *8MW*/*Na^3^* was used to calculate the X-ray density (*ρ_x-ray_*), where *MW* stands for the sample’s molecular weight, *N* for Avogadro’s number and ‘*a*’ for the lattice parameter. Table 2 makes it quite evident that when component x increases, the *ρ_x-ray_* also increases. The increase in *ρ_x-ray_* is attributable to the higher atomic weight of the Dy^3+^ ions compared to Fe^3+^ ions which has the lower atomic mass. The percentage porosity (*P*) was obtained via; porosity = (*ρ_x-ray_* − *ρ_exp_*)/*ρ_x-ray_*. *P* increased with the increasing amount of Dy^3+^ (Table 2).

X-ray diffraction pattern analysis was carried out to estimate the cation distribution of all the synthesized samples. The cation distribution was estimated using the Bertaut method [37]. In this technique, a few reflection pairings were chosen based on the expression:(8)$$\frac{I_{hkl}^{Obs.}}{I_{h^{\prime }k^{\prime }l^{\prime }}^{Obs.}}=\frac{I_{hkl}^{Cal.}}{I_{h^{\prime }k^{\prime }l^{\prime }}^{Cal.}}$$
where $I_{hkl}^{Obs.}$ and $I_{hkl}^{Calc.}$ are the observed and calculated intensities for reflection (hkl), respectively. The comparison of calculated and experimentally observed intensity ratios for reflections provides the most accurate information on cation distribution. In the present work, the intensity ratio was calculated using the reflections of (440), (400) and (220) planes. These planes are thought to be cation distribution sensitive [38,39,40]. The intensity computation is unaffected by the temperature and absorption variables, thus we do not account for these in our calculations.
(9)$$R=\left(\frac{I_{hkl}^{Obs.}}{I_{h^{\prime }k^{\prime }l^{\prime }}^{Obs.}}\right)-\left(\frac{I_{hkl}^{Cal.}}{I_{h^{\prime }k^{\prime }l^{\prime }}^{Cal.}}\right)$$

These intensities are almost completely unrelated to the oxygen vacancies. The computations were performed for several cation combinations at A- and B-sites. The following formula can be used to determine the relative integrated intensity of a certain diffraction line from powder specimens:(10)$$I_{hkl}={\left|F\right|}_{hkl}^{2}P\cdot L_{P}$$
where ‘*P*’ and ‘*F*’ are the multiplicity and structure factors, respectively. *L_P_* the Lorentz polarization factor is given as:(11)$$L_{P}=\frac{1+{cos}^{2}2\theta }{{sin}^{2}{cos}^{2}2\theta }$$

Values of the atomic scattering factor for different ions were obtained from the literature [32]. Table 3 displays the final cation distribution that was estimated through above mentioned details. Dy^3+^ and Co^2+^ ions occupied B-site only. Cr^3+^ and Fe^3+^ ions are distributed over both A and B crystallographic sites, where 60 and 40% of Cr^3+^ ions distributed over A- and B-sites, respectively (Table 3). Estimated results are fairly matches with the hypothesis of Smit and Wijn [41].

The average ionic radii of the tetrahedral (A) and octahedral [B] sites, *r_A_* and *r_B_*, respectively, were obtained using the relations given in literature [42]. The *r_A_* remain constant whereas *r_B_* increased with the increasing Dy^3+^ content x (Table 3). The variation in *r_A_* and *r_B_* is related to the relative occupancies of Co^2+^, Cr^3+^, Fe^3+^ and Dy^3+^ cations having their distinct ionic radius of 0.63 Å, 0.72 Å, 0.67 Å and 0.99 Å, respectively. The theoretical lattice parameter (*a_th_*) was obtained using the equation discussed elsewhere [43]. The values of *a_th_* lie between 8.755 Å and 8.803 Å (Table 3) that agreed well with ‘*a*’ and ‘*a*_0_’.

Oxygen positional parameter or anion parameter (u) is the distance between the oxygen ion and the face of the cube edge along the cube diagonal of the spinel lattice. The oxygen positional parameter ‘*u*’ was calculated using the relation discussed elsewhere [44]. Table 3 display that ‘u’ values are ranging from 0.3858 to 0.3854 Å. The oxygen ions appear to be bigger compared to the Cr^3+^, Co^2+^, Dy^3+^ and Fe^3+^ metallic ions in the majority of oxide spinels. In general, ‘*u*’ has a value in the vicinity of 0.375 Å in spinel-like structures, where of O^2-^ ions are exactly arranged in cubic-closed packing. However, in real spinel lattice, this ideal pattern is marginally distorted. The data for the oxygen positional parameter and the lattice constant provide some information about the lattice distortion, in particular about the angular distortion of the oxygen octahedral.

SEM images of all the samples in the powder form are shown in Figure 5. Surface morphology as examined from the SEM images show a porous structure of all the prepared samples, which is related to the nanoparticle nature of the prepared samples. Moreover, the sintering temperature is not high to make the samples with dense structure, particularly when the samples were prepared by the sol-gel method. Energy dispersive analysis of X-ray (EDAX) (Figure 6) and colour elemental mapping (Figure 7) were carried out to investigate the elemental stoichiometry of the prepared samples. Figure 8 shows the HRTEM images, particle size distribution and selected area electron diffraction (SAED) patterns. The average particle size (t) acquired from the TEM analysis is in the range of 30–50 nm which is higher compared to the crystallite size analysed from the XRD and W-H analysis. SAED patterns measurements confirmed the cubic spinel structure of the samples and the results matches well with the XRD data.

### 3.2. Spectral Aspects

Local symmetry and ordering phenomenon in crystalline solids can be identified using the infrared spectroscopy [45]. The cubic spinel exhibits four infrared active vibrations due to its space group $Fd3mO_{h}^{7}$. The oxygen bond-strength and bond-length, and dimensions of the unit cell are the governing factors that are altered by the substitution of various ions. The infrared spectrum is dependent by the nature of each of these factors. IR absorption spectra captured in the range of 300–800 cm^−1^ are displayed in Figure 9. Vibrational band frequency (*ν*_1_ and *ν*_2_) obtained from IR spectra are given in Table 4. The first lower frequency band *ν*_1_ is observed in the range of 379−409 cm^−1^, whereas band *ν*_2_ is observed in the range of 564–587 cm^−1^. These bands are ascribed to the Fe^3+^–O bonds on the A- and B-site, respectively [45]. The change in *ν*_2_ band position to higher frequency side is related to the occupancy of Dy^3+^ cations at B-site.

Force constants for A- and B-sites represented as *K_T_* and *K_O_*, respectively, were obtained using the relations discussed elsewhere [46].

Average force constant (*K_av_*) was computed from the average of *K_T_* and *K_O_* (Table 4). The stiffness constant (*C*_11_), bulk modulus (*B*), rigidity modulus (*G*), Young’s modulus (*E*), mean wave velocity (*V_m_*), transverse wave velocity (*V_t_*), longitudinal wave velocity (*V_l_*), Poisson’s ratio (*σ*) were obtained using the relations [47]:(12)$$C_{11}=\frac{K_{av}}{a}$$
(13)$$G=\rho v_{t}^{2}$$
(14)E=(1+σ)2G
(15)$$B=\frac{1}{3}\left(C_{11}+2C_{12}\right)$$
(16)$$v_{1}={\left(\frac{C_{11}}{\rho }\right)}^{1/2}$$
(17)$$V_{t}=\frac{V_{1}}{3}$$
(18)$$V_{m}={\left[\frac{1}{3}\left(\frac{2}{V_{t}^{3}}+\left(\frac{1}{V_{1}^{3}}\right)\right)\right]}^{-1/3}$$

Table 4 display the values of the different elastic parameters derived using the relations (12)–(18). The values of elastic properties of Co-Cr ferrite were decreased by the addition of Dy^3+^ ions. Rendering to isotropic elasticity theory, Poisson’s ratio should be in the range of −1 to 0.5 for the materials to exhibit acceptable elastic behaviour [45,48,49,50]. Herein, obtained Poisson’s ratio is ~0.35 suggesting the reasonable elastic behaviour of the CoCr_0.5_Dy_x_Fe_1.5−x_O_4_. Elastic moduli were described by the strength of the ionic bonds; stronger bonds increase the elastic moduli, and vice versa. It is evident that replacement of Fe^3+^ ions by Dy^3+^ ions weakens the interatomic bonding and lowers the elastic modulus values. Debye temperature (*θ_D_*) was obtained using the Waldron’ relation [45]:(19)$$\theta_{D}=\frac{\hbar c}{k_{B}}\upsilon_{av}=1.438\times \upsilon_{av}$$
where $v_{av}=\left(v_{1}+v_{2}\right)/2$. Substitution of Dy^3+^ for Fe^3+^ ions increase the *θ_D_* from 540 to 565 K (Table 4). The increase in *θ_D_* could be related to the increment in the P-type conduction in the CoCr_0.5_Dy_x_Fe_1.5−x_O_4_ [51].

The corrected porosity (zero porosity) was obtained by putting the values of porosity fraction (*f*) in the following equations [48,49,50]:(20)$$E_{0}={\left[\frac{1}{E}\left\{1-\frac{3f\left(1-\sigma \right)\left(9+5\sigma \right)}{2\left(7-5\sigma \right)}\right\}\right]}^{-1}$$
(21)$$G_{0}={\left[\frac{1}{G}\left\{1-\frac{15f\left(1-\sigma \right)}{\left(7-5\sigma \right)}\right\}\right]}^{-1}$$
(22)$$B_{0}=\frac{E_{0}G_{0}}{3(3E_{0}{-G}_{0})}$$
(23)$$\sigma_{0}=\frac{E_{0}}{2G_{0}}-1$$

Table 5 indicating the increasing nature of *E*_0_, *G*_0_, *B*_0_ and *σ*_0_ elastic parameters with the substitution of Dy^3+^ ions in Co-Cr ferrites. Strengthening the bonding among the corresponding ions with the Dy^3+^ ion substitution suggesting the increasing elastic behaviour of the Co-Cr ferrite.

### 3.3. Magnetic Aspects

The magnetization (*M*) vs. applied magnetic field (*H*) curves are shown in Figure 10. The Dy^3+^ substitution alters the magnetic behaviour, as seen by the magnetization curve. A maximum magnetic field of ±15 kOe was applied for the M-H curve measurements. Magnetization not completely saturated even at ±15 kOe. Table 6 listed the magnetization at 15 kOe (*Ms*), remanent magnetization (*R*) and coercivity (*Hc*) obtained from the magnetization curves. Ms increased, whereas Hc dropped with the replacement of Fe^3+^ by Dy^3+^ ions.

The cation distribution at the A and B sites affect the *Ms* according to Neel’s two sublattice collinear spin model. The theoretical magnetic moment (*n_B_*) is expressed as $n_{B}^{N}=M_{B}-M_{A},$ where *M_B_* and *M_A_* are the *B* and *A* sublattice magnetic moments in *μ_B_*, respectively. Increase in *Ms* with Dy^3+^ substitution is related to the magnetic moments of the constituent ions in CoCr_0.5_Dy_x_Fe_1.5−x_O_4_, where Cr^3+^ (3 *μ_B_*), Fe^3+^ (5 *μ_B_*), Dy^3+^ (10.5 *μ_B_*) and Co^2+^ (3 *μ_B_*) all have known magnetic moments. The introduction of Dy^3+^ ions that replacing Fe^3+^ ions at the B-site is responsible for the enhancement in the magnetic moment of B-sub lattice, whereas the magnetic moment of A-sublattice remain unchanged, thus the total magnetic moment is increased.

The experimental magnetic moment (*n_B_*) was obtained experimentally using the relation discussed elsewhere [52]. Values of experimentally and theoretically observed magnetic moments are given in Table 6. One of the typical magnetic parameter of the magnetic material is the remanent ratio *R* = *Mr*/*Ms*. It is a sign of how easily the direction of magnetization reorients to the closest easy axis of magnetization direction once the magnetic field is withdrawn. It has been noted that the R values fall between 0.418 and 0.490 and exhibit an upward trend with the Dy^3+^ substitution. Table 6 makes it abundantly evident that the Dy^3+^ substitution caused the coercivity (*H_C_*) to drop from 504 Oe to 387 Oe. The decrease in coercivity can be related to the Brown’s relation where the saturation magnetization is inversely proportional to coercivity [53]. The magnetocrystalline anisotropy constant (*K*_1_) obtained using the values of *Hc* and *Ms* is given in Table 6.

The change in magnetization behaviour can be understood based on the magnetic interactions [53,54]. The super-exchange process makes it feasible for the cations to interact magnetically via the intermediary oxygen ions in three distinct ways, known as *A*-*A* interaction, *B*-*B* interaction and *A*-*B* interaction (Figure 11). *A* and *B* represents the cations at tetrahedral and octahedral site, respectively. The cation distribution is listed in Table 3 referring that A = Cr and Fe ions, whereas B = Co, Dy, Cr and Fe ions. The distance of two magnetic ions (*M^I^* and *M^II^*) from oxygen ions, and the angle among *M^I^*–O–*M^II^* represented as *θ*, are critical parameters determining the interaction energies between the two magnetic ions. Maximum interaction energy is produced when the cations are angled at around *θ =* 180°. As the distance among the cations and the oxygen anion increases, the exchange energy rapidly reduces. The *A*-*B* interaction is the strongest among these three interactions, and as a result, the lengths of the cation and anion bonds are smaller. The inter-ionic lengths among the cations (*Me*–*Me*), cation and anion (*Me*–*O*) as well as bond angles between the cations and cation-anion were acquired using the equations tabulated in Table 7.

Table 8 shows that the Dy^3+^ substitution has increased the *Me–O* cation-anion and *Me–Me* cation-cation lengths. The *BO_6_* octahedra bulges caused by the bigger Dy^3+^ ions replacing the smaller Fe^3+^ ions, which enhanced the *B–O* bond length. The inter-cation lengths increased as a result of tetrahedra–*AO_4_* sinking instead of altering the $\bar{4}3m$ symmetry. This may have increased the anion-anion (*O–O*) length and cation-cation lengths. The Dy^3+^ substitution resulted into the change in bond angles, where; *θ_5_* belonging to the *A*-*A* interaction marginally increased, *θ_3_* and *θ_4_* referring to the *B*-*B* interaction decreased, *θ*_1_and *θ*_2_ pertaining to the *A*-*B* interaction increased.

These findings imply that the *A*-*B* exchange-interaction in the Co-Cr ferrite with the Dy^3+^ substitution is stronger than the *A*-*A* and *B*-*B* interactions, and that such an increment in the exchange interaction may improve the saturation magnetization values.

### 3.4. Magnetoelectric Properties

It is observed from the magnetic measurements that the substitution of Dy in Co-Cr ferrite increases the saturation magnetization; however, coercivity decreased considerably. Thus, three compositions of ferrite CoCr_0.5_Fe_1.5_O_4_, CoCr_0.5_Dy_0.05_Fe_1.45_O_4_ and CoCr_0.5_Dy_0.1_Fe_1.4_O_4_ were chosen as ferromagnetic component phase for its nanocomposite with ferroelectric BaTiO_3_. Dynamic method [55] was used to measure the magneto-electric (ME) coefficient ($\alpha_{ME}$) of these samples. Detailed procedure is given in one of our earlier reports [56]. Plots of $\alpha_{ME}$ with dc magnetic field (*H_DC_*) is shown in Figure 12. $\alpha_{ME}$ initially rapidly increased with *H_DC_*~1000 Oe. Magnetostriction in ferromagentic_-_ferrite phase increased with the increase with *H_DC_* and reaches near to saturation at a certain applied magnetic field [57]. Therefore, the strain introduced in ferromagnetic-ferrite phase induced the electric field in the BaTiO_3_ ferroelectric phase. This mechanism introduces and varies the ME voltage in the CoCr_0.5_Dy_x_Fe_1.5−x_O_4_ + BaTiO_3_ composites. Maximum $\alpha_{ME}$ is observed in CoCr_0.5_Dy_0.05_Fe_1.45_O_4_ + BaTiO_3_ compared to other samples which is related to the magnetic properties of CoCr_0.5_Dy_0.05_Fe_1.45_O_4_ compound.

## 4. Conclusions

The Rietveld refined XRD patterns showed the formation of single-phase cubic spinel structure of CoCr_0.5_Dy_x_Fe_1.5−x_O_4_. No traces of any secondary and impurity phases were developed with the substitution of Dy^3+^. Such observations are well-supported by the XRD and SAED patterns. Dy^3+^ substitution induced the tensile strain in the Co-Cr ferrite. The SEM and TEM confirmed the nanocrystalline nature of all the samples. The cation distribution data suggest that the Dy^3+^ and Co^2+^ ions occupied B-site, whereas Cr^3+^ and Fe^3+^ ions have the occupancy at A- and B-sites. The two characteristics absorption bands related to the spinel ferrite were observed in IR spectra. Elastic parameters obtained from the IR data were influenced with the Dy^3+^ substitution. The magnetization of Co-Cr ferrite increased with the substitution of Dy^3+^ ions and is related to the strengthening of A-B interactions. Overall, the magnetic properties of the Co-Cr ferrite is greatly enhanced by the substitution of rare-earth Dy ions. Increased saturation magnetization and coercivity makes the materials suitable for ME composite as a ferromagnetic-magnetostrictive counterpart. Nanocomposite of ferromagnetic CoCr_0.5_Dy_0.05_Fe_1.45_O_4_ with ferroelectric BaTiO_3_ showed a significant magnetoelectric coupling. The considerable ME coupling of CoCr_0.5_Dy_0.05_Fe_1.45_O_4_ + BaTiO_3_ makes it a potential candidate for its application in ultra-low power and highly dense logic-memory, electronic, sensors, etc.

## Figures and Tables

**Figure 1 nanomaterials-13-01165-f001:**
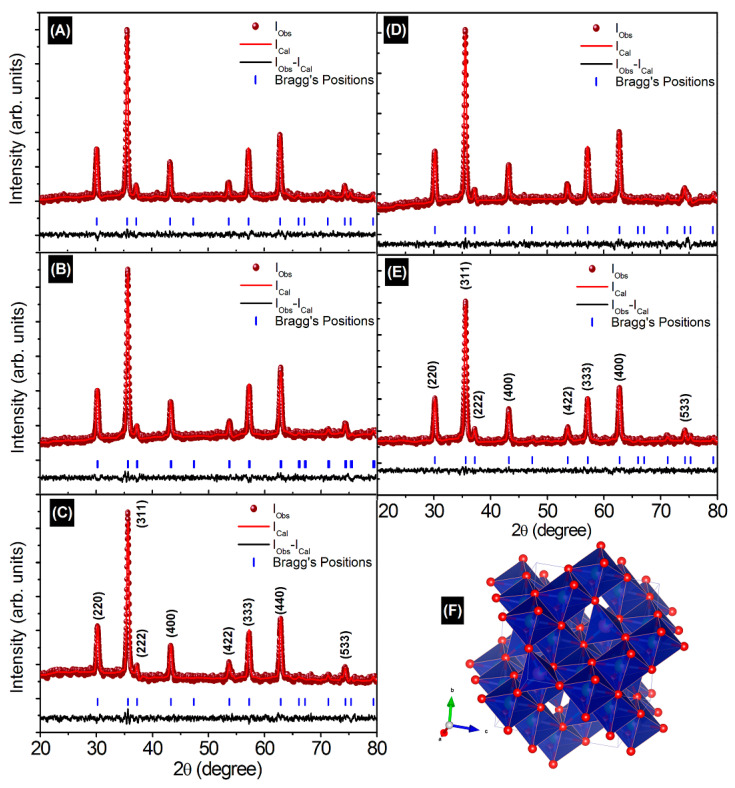
Rietveld refined XRD patterns of (**A**) x = 0.0, (**B**) x = 0.025, (**C**) x = 0.050, (**D**) x = 0.075, (**E**) x = 0.1 and (**F**) structure of CoCr_0.5_Dy_x_Fe_1.5−x_O_4_.

**Figure 2 nanomaterials-13-01165-f002:**
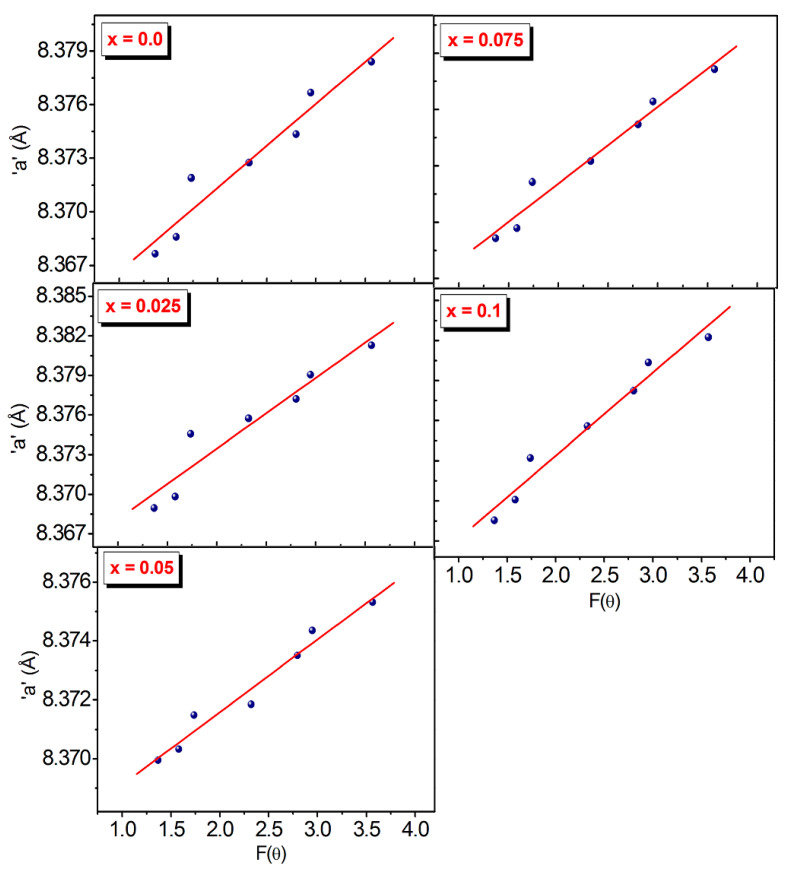
Plots of NR function for CoCr_0.5_Dy_x_Fe_1.5−x_O_4_.

**Figure 3 nanomaterials-13-01165-f003:**
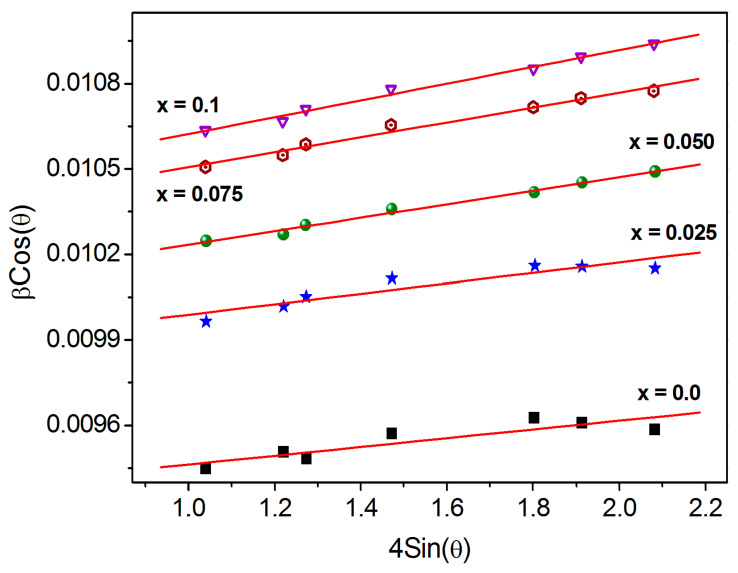
W-H plots of CoCr_0.5_Dy_x_Fe_1.5−x_O_4_.

**Figure 4 nanomaterials-13-01165-f004:**
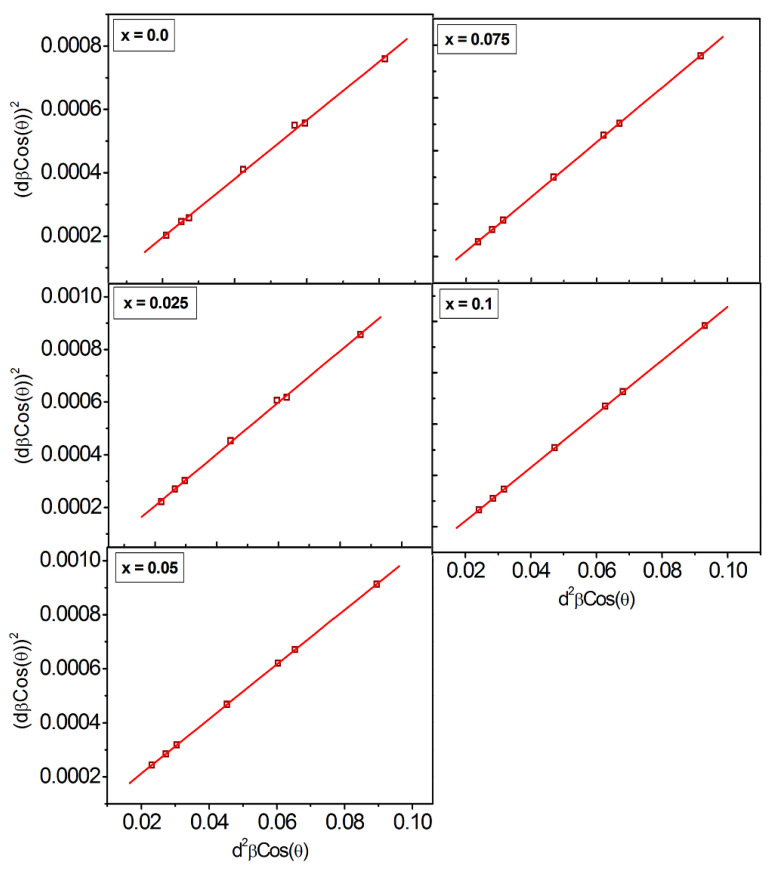
Size—strain plots of CoCr_0.5_Dy_x_Fe_1.5−x_O_4_.

**Figure 5 nanomaterials-13-01165-f005:**
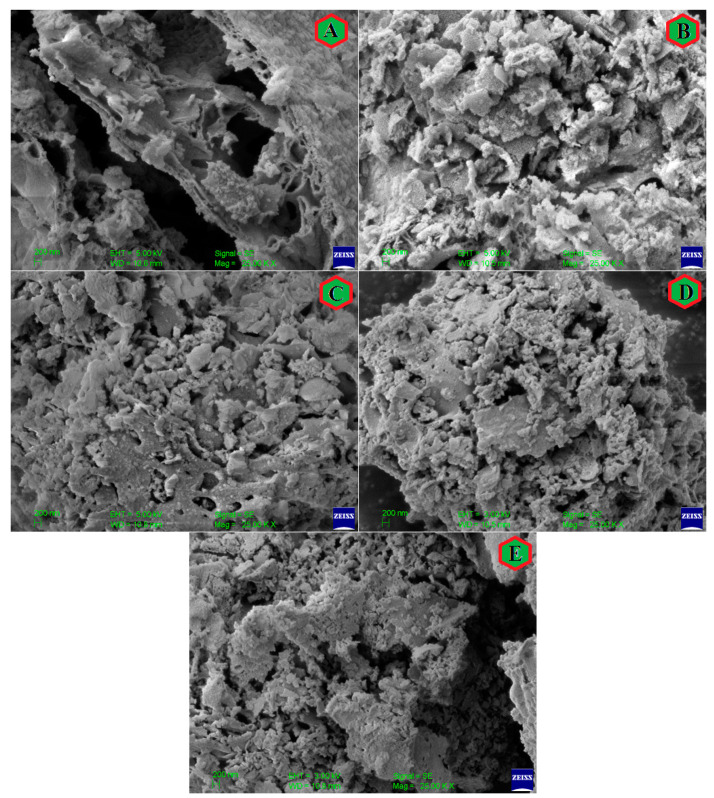
SEM images of CoCr_0.5_Dy_x_Fe_1.5−x_O_4_ where (**A**) x = 0.0, (**B**) x = 0.025, (**C**) x = 0.050, (**D**) x = 0.075 and (**E**) x = 0.1.

**Figure 6 nanomaterials-13-01165-f006:**
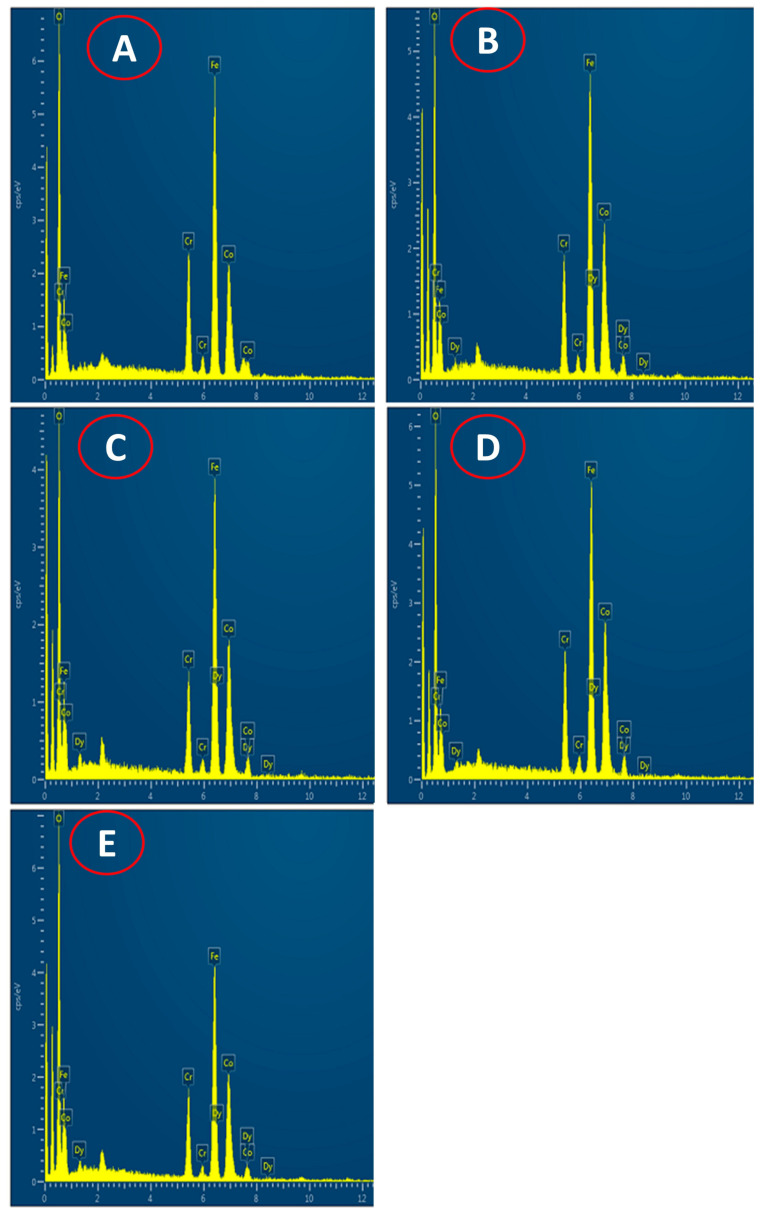
EDAX spectra of CoCr_0.5_Dy_x_Fe_1.5−x_O_4_ where (**A**) x = 0.0, (**B**) x = 0.025, (**C**) x = 0.050, (**D**) x = 0.075 and (**E**) x = 0.1.

**Figure 7 nanomaterials-13-01165-f007:**
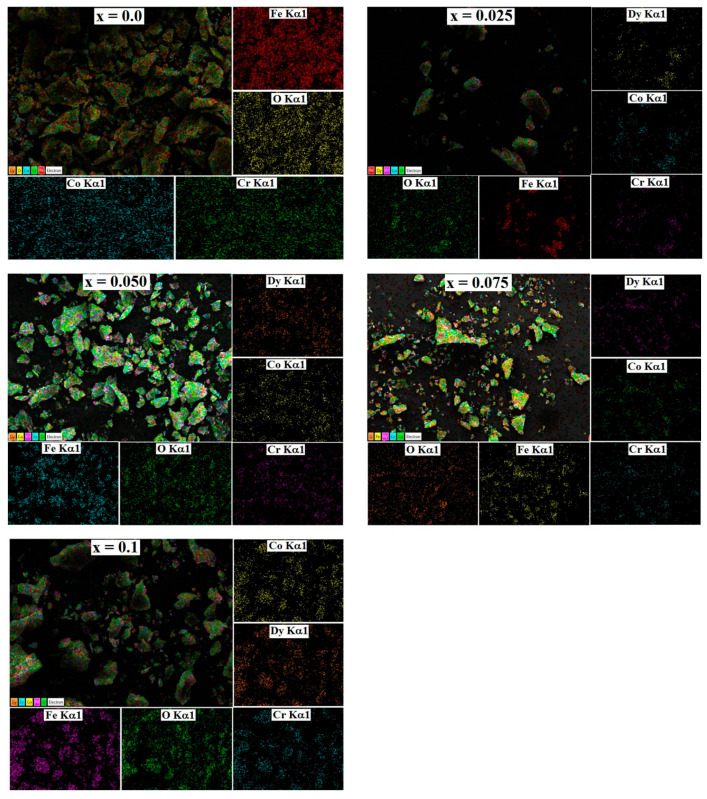
Elemental colour mapping of all the samples of CoCr_0.5_Dy_x_Fe_1.5−x_O_4_ where x = 0.0, 0.025, 0.05, 0.075 and 0.1.

**Figure 8 nanomaterials-13-01165-f008:**
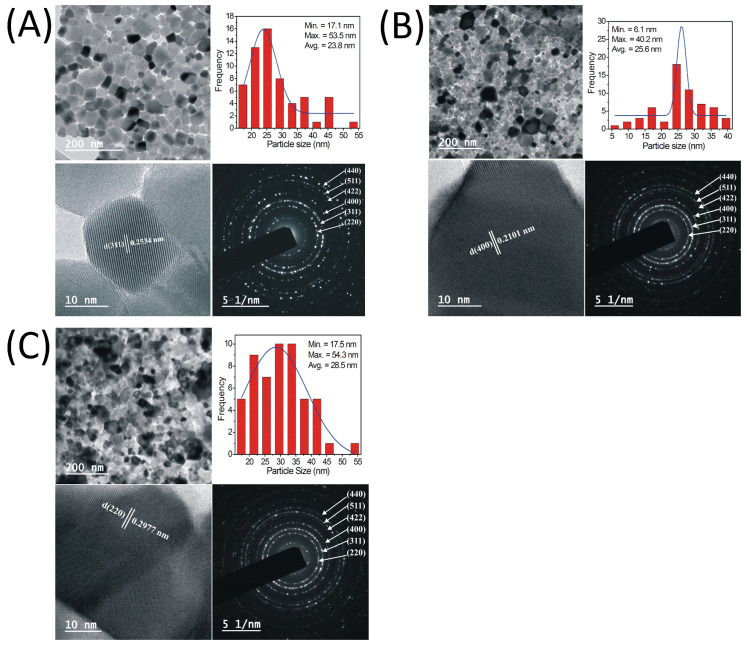
TEM, HRTEM, particle size distribution and SAED patterns of CoCr_0.5_Dy_x_Fe_1.5−x_O_4_ for (**A**) x = 0.0, (**B**) x = 0.05 and (**C**) x = 0.1.

**Figure 9 nanomaterials-13-01165-f009:**
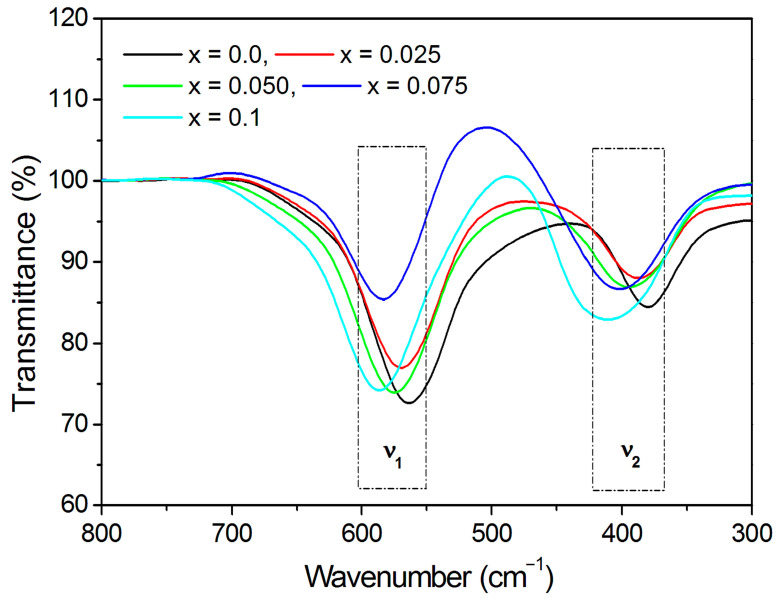
FTIR spectra of CoCr_0.5_Dy_x_Fe_1.5−x_O_4_.

**Figure 10 nanomaterials-13-01165-f010:**
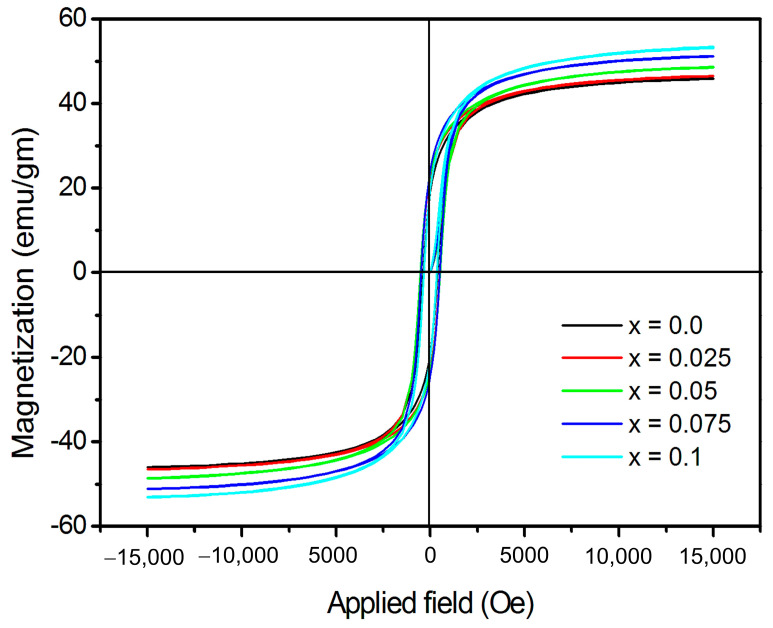
M-H hysteresis loops of CoCr_0.5_Dy_x_Fe_1.5−x_O_4_.

**Figure 11 nanomaterials-13-01165-f011:**
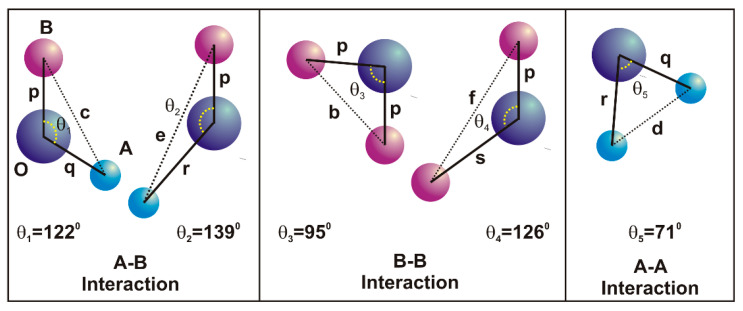
A-B, B-B and A-A interactions of CoCr_0.5_Dy_x_Fe_1.5−x_O_4_.

**Figure 12 nanomaterials-13-01165-f012:**
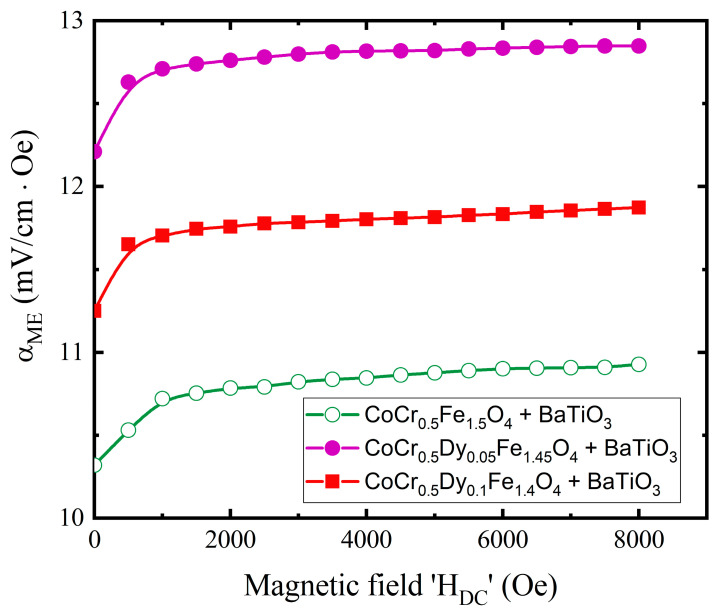
Variation of magnetoelectric voltage coefficient (*α_ME_*) with DC magnetic field (*H_DC_*) for three compositions of ferrite + BaTiO_3_ nanocomposites.

**Table 1 nanomaterials-13-01165-t001:** Rietveld Refined parameters of CoCr_0.5_Dy_x_Fe_1.5−x_O_4_.

‘x’	‘a’ (Å)	R_P_	R_WP_	R_Exp_	χ^2^
0.0	8.3626	43.9	17.4	17.32	1.01
0.025	8.3652	39.5	18.8	16.53	1.29
0.050	8.3671	43.4	22.1	19.69	1.26
0.075	8.3687	41.6	20.5	18.33	1.25
0.1	8.3717	42.7	18.4	18.77	1.02

**Table 2 nanomaterials-13-01165-t002:** Lattice parameter obtained from N-R function (*a*_0_), X-ray density (*ρ_x-ray_*), bulk density (*ρ_exp_*), porosity (*P*), specific surface area (*S*), average crystallite size (*t_xrd_*), lattice strain (*ε*) for the series CoCr_0.5_Dy_x_Fe_1.5−x_O_4_. Figures in the square bracket refers to the estimated error.

‘*x*’	N-R Function ‘*a*_0′_ (Å) [0.002°]	‘ρ_*x*-*ray*_’ (g/cc) [0.005]	‘*ρ_exp_*’ (g/cc) [0.005]	‘*P*’ (%) [0.1]	‘*S*’ (m^2^/g) [2]	‘*t_xrd_*’ (nm) [0.1]	‘*t_W-H_*’ (nm) [0.1]	*ε* × 10^−4^ [0.05 × 10^−4^]	‘*t_SSP_*’ (nm) [0.1]	*ε* × 10^−3^ [0.05 × 10^−4^]
0.0	8.3619	5.267	4.92	6.63	84	14.5	14.9	1.54	15.0	6.47
0.025	8.3647	5.324	4.90	8.08	88	13.9	14.1	1.83	14.2	6.70
0.050	8.3671	5.376	4.89	9.11	91	13.5	13.9	2.37	13.7	7.18
0.075	8.3715	5.430	4.88	10.17	94	13.1	13.5	2.61	13.4	7.42
0.1	8.3752	5.483	4.84	11.65	97	12.8	13.4	2.96	13.3	7.64

**Table 3 nanomaterials-13-01165-t003:** Cation distribution of A- and B-sites, mean ionic radii (*r_A_* and *r_B_*), oxygen parameter (*u*), theoretical lattice constant (*a_th_*) for CoCr_0.5_Dy_x_Fe_1.5−x_O_4_. Figures in the square bracket refers to the estimated error.

x	A-Site	B-Site	‘*r_A_*’ (Å) [0.001]	‘*r_B_*’ (Å) [0.001]	‘*u*’ (Å) [0.001]	‘*a_th_*’ (Å) [0.002]
0.0	(Cr_0.3_Fe_0.7_)	[Co_1.0_Cr_0.2_Fe_0.8_]	0.710	0.792	0.3858	8.755
0.025	(Cr_0.3_Fe_0.7_)	[Co_1.0_Dy_0.025_ Cr_0.2_Fe_0.775_]	0.710	0.796	0.3857	8.767
0.050	(Cr_0.3_Fe_0.7_)	[Co_1.0_Dy_0.05_ Cr_0.2_Fe_0.75_]	0.710	0.801	0.3856	8.779
0.075	(Cr_0.3_Fe_0.7_)	[Co_1.0_Dy_0.075_ Cr_0.2_Fe_0.725_]	0.710	0.805	0.3855	8.791
0.1	(Cr_0.3_Fe_0.7_)	[Co_1.0_Dy_0.1_ Cr_0.2_Fe_0.7_]	0.710	0.810	0.3854	8.803

**Table 4 nanomaterials-13-01165-t004:** Elastic properties of CoCr_0.5_Dy_x_Fe_1.5−x_O_4_. Absorption bands (*ν*_1_ and *ν*_2_), molecular weights at tetrahedral A-site (*M_A_*) and octahedral B-site (*M_B_*), force constants (*K_T_* and *K_O_*), the stiffness constant (*C*_11_), bulk modulus (*B*), rigidity modulus (*G*), Young’s modulus (*E*), mean wave velocity (*V_m_*), transverse wave velocity (*V_t_*), longitudinal wave velocity (*V_l_*), Poisson’s ratio (*σ*) and Debye temperatures by Waldron (*Θ*_D_) method. 2% of error is consider and estimated in *ν*_1_ and *ν*_2_ values, thus approximately similar percentage of error should be considered in all the derived parameters.

‘x’	0.0	0.025	0.050	0.075	0.1
*ν*_1_ (cm^−1^)	564.95	569.03	575.15	581.26	587.38
*ν*_2_ (cm^−1^)	379.35	387.51	395.67	401.78	409.74
*M_A_*	78.69	78.69	78.69	78.69	78.69
*M_B_*	154.01	156.67	159.34	162.01	164.67
*K_T_* (×10^5^ dynes/cm)	191.38	194.15	198.35	202.59	206.88
*K_O_* (×10^5^ dynes/cm)	117.68	124.93	132.46	138.87	146.95
*K_av_* (×10^5^ dynes/cm)	154.5	159.5	165.4	170.7	176.9
*C*_11_ (GPa)	184.57	190.51	197.42	203.69	210.98
*B* (GPa)	184.57	190.51	197.42	203.69	210.98
*G* (GPa)	61.52	63.50	65.81	67.90	70.33
*E* (GPa)	166.11	171.46	177.68	183.32	189.88
*V_m_* (m/s)	3794	3834	3884	3926	3976
*V_t_* (m/s)	3418	3454	3499	3536	3581
*V_l_* (m/s)	5920	5982	6060	6125	6203
*σ*	0.35	0.35	0.35	0.35	0.35
*θ_D_* (K)	540	545	552	558	565

**Table 5 nanomaterials-13-01165-t005:** Pore free elastic properties of CoCr_0.5_Dy_x_Fe_1.5−x_O_4_.

Com. ‘x’	*E*_0_ (GPa)	*G*_0_ (GPa)	*B*_0_ (GPa)	*σ* _0_
0.0	197.87	72.32	249.86	0.368
0.025	205.91	75.21	261.82	0.369
0.050	217.28	79.24	280.71	0.371
0.075	228.36	83.15	300.03	0.373
0.1	239.03	86.96	317.14	0.374

**Table 6 nanomaterials-13-01165-t006:** Magnetic properties of CoCr_0.5_Dy_x_Fe_1.5−x_O_4_. Figures in the square bracket refers to the estimated error.

Com. ‘x’	*Ms@15kOe* (emu/g) [0.5]	*Mr* (emu/g) [0.2]	*Hc* (Oe) [20]	*R* [0.05]	*K*_1_ × 10^4^ [0.05 × 10^4^]	*n_B_* (μ_B_) [0.02]	*nB_N_* (μ_B_) [0.02]
0.0	45.9	19.2	504	0.418	2.41	1.912	3.200
0.025	46.5	21.4	478	0.460	2.31	1.960	3.337
0.050	48.6	22.8	433	0.469	2.19	2.071	3.474
0.075	50.3	23.9	409	0.475	2.14	2.168	3.611
0.1	50.4	25.7	387	0.490	2.11	2.283	3.748

**Table 7 nanomaterials-13-01165-t007:** Equations to obtain the cation-anion (*Me–O*) and cation-cation (*Me–Me*) lengths and bond angles. Here ‘a’ is the lattice constant and ‘u’ is the oxygen positional parameter.

Me–O	Me–Me	Bond Angles
$p=a\left(\frac{1}{2}-u^{3m}\right)$	$b=\sqrt{2}\left(\frac{a}{4}\right)$	$\theta_{1}={cos}^{-1}\left(\frac{p^{2}+q^{2}-c^{2}}{2pq}\right)$
$q=a\sqrt{3}\left(u^{3m}-\frac{1}{8}\right)$	$c=\sqrt{11}\left(\frac{a}{11}\right)$	$\theta_{2}={cos}^{-1}\left(\frac{p^{2}+r^{2}-e^{2}}{2pr}\right)$
$r=a\sqrt{11}\left(u^{3m}-\frac{1}{8}\right)$	$d=\sqrt{3}\left(\frac{a}{4}\right)$	$\theta_{3}={cos}^{-1}\left(\frac{{2p}^{2}-b^{2}}{{2p}^{2}}\right)$
$s=a\sqrt{3}\left(u^{3m}+\frac{1}{8}\right)$	$e=3\sqrt{3}\left(\frac{a}{8}\right)$	$\theta_{4}={cos}^{-1}\left(\frac{p^{2}+s^{2}-f^{2}}{2ps}\right)$
	$f=\sqrt{6}\left(\frac{a}{4}\right)$	$\theta_{5}={cos}^{-1}\left(\frac{r^{2}+q^{2}-d^{2}}{2rq}\right)$

**Table 8 nanomaterials-13-01165-t008:** Inter-ionic lengths (Me–O, Me–Me) and bond angles (θ) of CoCr_0.5_Dy_x_Fe_1.5−x_O_4_. Figures in the square bracket refers to the estimated error.

↓Formula\‘x’→	0.0	0.025	0.050	0.075	0.1
**Me–O lengths (nm)** [0.002]					
‘*p*’	0.2003	0.2004	0.2006	0.2007	0.2009
‘*q*’	0.1968	0.1968	0.1968	0.1968	0.1968
‘*r*’	0.3768	0.3768	0.3767	0.3767	0.3766
‘*s*’	0.3677	0.3678	0.3679	0.3680	0.3681
**Me–Me lengths (nm)** [0.002]					
‘*b*’	0.2960	0.2961	0.2962	0.2963	0.2965
‘*c*’	0.3471	0.3472	0.3474	0.3475	0.3476
‘*d*’	0.3625	0.3626	0.3628	0.3629	0.3631
‘*e*’	0.5438	0.5439	0.5442	0.5444	0.5446
‘*f*’	0.5127	0.5128	0.5131	0.5133	0.5135
**Bond angles (°)** [0.05]					
*θ* _1_	122.12	122.17	122.23	122.25	122.28
*θ* _2_	139.24	139.29	139.32	139.34	139.37
*θ* _3_	95.84	95.79	95.77	95.71	95.68
*θ* _4_	126.42	126.39	126.38	126.37	126.35
*θ* _5_	71.04	71.05	71.06	71.07	71.07

## Data Availability

The data presented in this study are available on request from the corresponding author.

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
