# Peer review of "A thorough Investigation of Rare-Earth Dy3+ Substituted Cobalt-Chromium Ferrite and Its Magnetoelectric Nanocomposite"

_nanomaterials, 2023, doi:10.3390/nano13071165_

Round 1

Reviewer 1 Report

This article investigates the effect of the addition of Dy3+ cations on the structure and properties of ferrite-chromites with the spinel structure CoCr0.5DyxFe1.5-xO4. The samples were successfully synthesized by the sol-gel method and structural, morphological and magnetic properties were studied for them by X-ray diffraction (XRD), infra-red spectroscopy (IR), scanning electron microscopy, transmission electron microscopy and vibrating sample magnetometer. A detailed analysis of the sizes of crystallites, microstresses and distortions of the crystal lattice of spinel in the presence of a dopant was carried out, which is an interesting section of this article, since such an analysis is rarely performed. The distribution of cations over the octahedral and tetrahedral positions of the spinel structure was determined, and the magnetic properties of the obtained solid solutions were interpreted taking this into account. The possibility of using CoCr0.5DyxFe1.5-xO4 as magnetostrictive phases of multiferroic ceramics piezoelectric - ferrite is demonstrated. The work is a complete study in which the characterization of the obtained products by various modern methods was carried out. There are several questions: 1. The authors use the citrate method to obtain CoCr0.5DyxFe1.5-xO4, by which products are obtained at 800 C. However, low-temperature synthesis is possible for spinel ferrites at lower temperatures (10.1016/j.jmmm.2015.08.084, 10.1134 /S0036023612040171). I think that the introduction should be supplemented with a short review of low-temperature methods for the synthesis of CoFe2O4 and solid solutions based on it. 2. The authors write: “The magneto-electric measurement was carried out by superimposing the sinusoidal magnetic field (HAC) produced by Helmholtz's coils with DC bias magnetic field (HDC) produced by electrically poled with applied electrified of 1.5 kv/cm at 200 ° C for 1 h.” Is there an error here? Possibly, the magnetoelectric measurements were carried out at room temperature with simultaneous application of constant and alternating magnetic fields on the samples, which were previously electrically polarized in the mode of 1.5 kv/cm at 200 °C for 1 h ? I think that the article can be published after a minor revision.

Author Response

Comment: This article investigates the effect of the addition of Dy3+ cations on the structure and properties of ferrite-chromites with the spinel structure CoCr0.5DyxFe1.5-xO4. The samples were successfully synthesized by the sol-gel method and structural, morphological and magnetic properties were studied for them by X-ray diffraction (XRD), infra-red spectroscopy (IR), scanning electron microscopy, transmission electron microscopy and vibrating sample magnetometer. A detailed analysis of the sizes of crystallites, microstresses and distortions of the crystal lattice of spinel in the presence of a dopant was carried out, which is an interesting section of this article, since such an analysis is rarely performed. The distribution of cations over the octahedral and tetrahedral positions of the spinel structure was determined, and the magnetic properties of the obtained solid solutions were interpreted taking this into account. The possibility of using CoCr0.5DyxFe1.5-xO4 as magnetostrictive phases of multiferroic ceramics piezoelectric - ferrite is demonstrated. The work is a complete study in which the characterization of the obtained products by various modern methods was carried out. 

Reply: We appreciate a very positive comment on our manuscript.

Comment #1: The authors use the citrate method to obtain CoCr0.5DyxFe1.5-xO4, by which products are obtained at 800 C. However, low-temperature synthesis is possible for spinel ferrites at lower temperatures (10.1016/j.jmmm.2015.08.084, 10.1134 /S0036023612040171). I think that the introduction should be supplemented with a short review of low-temperature methods for the synthesis of CoFe2O4 and solid solutions based on it. 

Reply: The as-prepared powders of all the samples were heat treated at 800 ºC for 4 h to get the final product. Low temperature is also suitable to prepare these kind of complex oxide system by chemical route. In general, literature reports suggest that 600 C for 4 h is also sufficient to crystallize the ferrite materials. However, in our case we have used 800 C for 4 h considering the incorporation of rare-earth Dy ions which may require higher energy (through temperature) to enter in the spinel lattice of cobalt ferrite.

We have now modified the introduction section with a small discussion supplemented with a short review of low-temperature methods for the synthesis of these material by incorporating the suggested references (Ref. 16 and Ref. 18) which are very helpful.

Comment #2:  The authors write: “The magneto-electric measurement was carried out by superimposing the sinusoidal magnetic field (HAC) produced by Helmholtz's coils with DC bias magnetic field (HDC) produced by electrically poled with applied electrified of 1.5 kv/cm at 200 ° C for 1 h.” Is there an error here? Possibly, the magnetoelectric measurements were carried out at room temperature with simultaneous application of constant and alternating magnetic fields on the samples, which were previously electrically polarized in the mode of 1.5 kv/cm at 200 °C for 1 h? I think that the article can be published after a minor revision.

Reply: “The magnetoelectric measurements were carried out at room temperature with simultaneous application of constant and alternating magnetic fields on the samples, which were previously electrically polarized in the mode of 1.5 kv/cm at 200 °C for 1 h” is more appropriate. Many thanks for the suggestion. We have made the necessary change in the manuscript.

Reviewer 2 Report

The magneto-electric particle composite is a very old topic that has attracted the attentions of the researchers due to the absoption/conversion abilities the EM waves.

In this manuscript, the authors woud like to compensate the loss of the magnetization in CoCrFeO due to the Cr3+ substitution. They propused that the Dy3+ can be substituted in the CoCrFeO system to improve the magneto-electric properties in CoCrFeO-BTO bi-phase system. I have a few comments for the authors:

Q1:  The resolution of the figures are terrible.  I cannot obtain enough information from those figures. The quality of all the figures must be improved.

Q2: The dynamic magnetostriction (magneto-electric) coefficients can be compared between the CoCrFeO(-BTO) and CoCrDyFeO(-BTO) systems in Fig 12.

Q3:  Some equations can be cancelled or add as refs. Not all the equations are necessary in this manuscript.

Q4:There are some problems in the ref list. For example, there is an error in ref 4, the author list need to be modified.

Q5:  It is suggested to perform some x-ray photoelectric spectroscopy experiments to classify the ionic valency. The results from the vibrational absorption spectrum cannot convince the readers, I think.

Author Response

Response to Reviewers Comments

Reviewer#2

Comment #1:  The resolution of the figures are terrible.  I cannot obtain enough information from those figures. The quality of all the figures must be improved.

Reply: Thanks for the suggestion. We have now improved the quality of figures.

Comment #2: The dynamic magnetostriction (magneto-electric) coefficients can be compared between the CoCrFeO(-BTO) and CoCrDyFeO(-BTO) systems in Fig 12.

Reply: We have measured the ME coefficients for CoCrFeO(-BTO) and CoCrDyFeO(-BTO) systems, however the response is lower for these systems compared to CoCr0.5Dy0.05Fe1.45O4 + BaTiO3.

Comment #3:  Some equations can be cancelled or add as refs. Not all the equations are necessary in this manuscript.

Reply: We have now removed some of the basic references from the manuscript.

Comment #4:There are some problems in the ref list. For example, there is an error in ref 4, the author list need to be modified.

Reply: Thanks for the suggestion. We have now revised the reference list and also modified the ref. 4.

Comment #5: It is suggested to perform some x-ray photoelectric spectroscopy experiments to classify the ionic valency. The results from the vibrational absorption spectrum cannot convince the readers, I think.

Reply: Indeed a good suggestion to perform the XPS measurements to confirm the ionic valency of the elements. However, we have not correlate and discussed the obtained results in the light of ionic valency. Rietveld refinement of XRD results presented in Fig. 1 clearly showed the formation of single phase cubic spinel structure with any signature of secondary phases. Thus, it is believed that all the elements in the CoCr0.5Dy0.05Fe1.45O4 ferrite system is maintained the stoichiometric proportion without violating the ionic valency of the starting materials. Moreover, we don’t have XPS facility at our end which will force us to send the samples elsewhere which could be time consuming.

Thus, we are extremely sorry that we could not do the XPS measurements and requesting reviewer to understand our situation.

Reviewer 3 Report

In the present work, authors reported the synthesis of CoCr0.5DyxFe1.5-xO4 ferrites by using sol-gel auto-combustion method, and then investigated their structural, morphological and magnetic properties. Results indicated that the magnetization of Co-Cr ferrite increased with the substitution of Dy3+ ions. This work has some reference function. However, some issues should be addressed.

1, The abstract can be polished and improved. The novelty problem statement described by the authors should be emphasized to attract general readers by providing more insights on the experimental observations. Also, the authors should elaborate the general applicability of the current work.

2, Authors may rearrange/polish the text and elaborated "Materials and Methods" section the way so anybody can repeat the procedures, like a recipe. If there is process flow diagram can be added in fig. 1, it would be helpful to non-specialist readers (optional). In addition, please provide all information of the methods of characterization such as XRD, SEM, etc. Example: Fourier transform infrared (FTIR) spectra of MgFe2O4 and MgFe2O4 /TiO2 samples were recorded on a Perkin-Elmer Spectrum 100 fitted with the universal attenuated total reflection (ATR) sampling accessory. The FTIR spectra recorded in absorbance units in the wavenumber range from 4000 to 400 cm-1 at a resolution of 4 cm-1. The reported spectra represent averages of 16 scans. Therefore, do this for all methods.

3, It was claimed that “Both “a0” and “a” have shown increment with the Dy3+ substitution which is an indication that Dy3+ entered in the Co-Cr ferrite spinel lattice and occupied the crystallographic sites.”. So why?

4, The images in this work were obscure. Please replace the blurry images by high-resolution ones.

5, Controllable and rational processing is a determinant in morphology of nanoparticles. How to improve the controllability and designability of nanoparticle preparation in this work? The authors should also pay attention to this challenge, and some pioneering and original researches about controllable fabrication of particles are suggested: Giant, 2021, 8, 100076; Journal of Materials Chemistry C, 2016, 4, 9738; Composites Part A, 2018, 115, 371.

6, It was said that “Surface morphology as examined from the SEM images show a porous structure of all the prepared samples.”. Why the aggregated particles showed a porous structure. It is an interesting phenomenon since very few literatures recorded this and authors may clarify the detail formation process.

7, In TEM results, the Co ferrites showed a typical polycrystalline structure. So, I am wondering how the average particle size acquired from the TEM analysis matched well with the crystallite size analyzed from the XRD and W-H analysis? Because only the particle size of monocrystals can match well with their crystalline size. Please give more details.

Author Response

Comment #1: The abstract can be polished and improved. The novelty problem statement described by the authors should be emphasized to attract general readers by providing more insights on the experimental observations. Also, the authors should elaborate the general applicability of the current work.

Reply: We authors appreciate the comment of the reviewer. Accordingly the abstract is polished and improved. Also the novelty problem statement is now emphasized.

Comment #2: Authors may rearrange/polish the text and elaborated "Materials and Methods" section the way so anybody can repeat the procedures, like a recipe. If there is process flow diagram can be added in fig. 1, it would be helpful to non-specialist readers (optional). In addition, please provide all information of the methods of characterization such as XRD, SEM, etc. Example: Fourier transform infrared (FTIR) spectra of MgFe2O4 and MgFe2O4 /TiO2 samples were recorded on a Perkin-Elmer Spectrum 100 fitted with the universal attenuated total reflection (ATR) sampling accessory. The FTIR spectra recorded in absorbance units in the wavenumber range from 4000 to 400 cm-1 at a resolution of 4 cm-1. The reported spectra represent averages of 16 scans. Therefore, do this for all methods.

Reply: We have now polished the text and elaborated “Materials and Methods” section. The flow diagram/chart of sol-gel method is available in may literature and thus we have not included in our manuscript. However, we have described the sample preparation in easy words and convincingly.

Comment #3:  It was claimed that “Both “a0” and “a” have shown increment with the Dy3+ substitution which is an indication that Dy3+ entered in the Co-Cr ferrite spinel lattice and occupied the crystallographic sites.”. So why?

Reply: It is known fact that the change in lattice constant is mainly related to the strain and substitution of foreign elements. We have discussed in the manuscript that “The rise in the lattice constant is mostly responsible for the increase in the tensile strain in Co-Cr ferrite after Dy substitution. The ionic radius of Dy3+ ions (0.99Å) is larger compared to Fe3+ ions (0.67Å) and thus brought expansion in crystallographic structure of Co-Cr ferrite”.

Comment #4:  The images in this work were obscure. Please replace the blurry images by high-resolution ones.

Reply: We have now improved the quality of images and replaced with high-resolution ones.

Comment #5:  Controllable and rational processing is a determinant in morphology of nanoparticles. How to improve the controllability and designability of nanoparticle preparation in this work? The authors should also pay attention to this challenge, and some pioneering and original researches about controllable fabrication of particles are suggested: Giant, 2021, 8, 100076; Journal of Materials Chemistry C, 2016, 4, 9738; Composites Part A, 2018, 115, 371.

Reply: We have now discussed the issue mentioned by the reviewer in the manuscript. Also included some pioneering and original literature in the manuscript (Ref. 14 and Ref. 15)

Comment #6:  It was said that “Surface morphology as examined from the SEM images show a porous structure of all the prepared samples.”. Why the aggregated particles showed a porous structure. It is an interesting phenomenon since very few literatures recorded this and authors may clarify the detail formation process.

Reply: The porous structure of the samples could be related to the nanoparticle nature of the prepared samples. Moreover, the sintering temperature is not high to make the samples with dense structure, particularly when the samples were prepared by the sol-gel method.

Comment #7:  In TEM results, the Co ferrites showed a typical polycrystalline structure. So, I am wondering how the average particle size acquired from the TEM analysis matched well with the crystallite size analyzed from the XRD and W-H analysis? Because only the particle size of monocrystals can match well with their crystalline size. Please give more details.

Reply: Indeed a good observation. We have now carefully analysed the particle size from the TEM. Accordingly the text in the manuscript is revised, “The average particle size (t) acquired from the TEM analysis is in the range of 30-50 nm which is higher compared to the crystallite size analyzed from the XRD and W-H analysis”

Reviewer 4 Report

Reviewers comments of manuscript, nanomaterials-2211303, A thorough investigation of rare-earth Dy3+ substituted cobalt-chromium ferrite and its magnetoelectric nanocomposite

The manuscript deals with the study of the structural, morphological and magnetic properties for the stoichiometric compositions of the ferrites, especially focusing on Dy3+ substituted cobalt-chromium ferrite results was presented in details in this MS. The quality of this MS probably validates its publication in Nanomaterials, but it is not acceptable for publication in its present form, there exists so many shortcoming and mistakes. However, seriously major revisions and corresponding carefully clarify a couple of issues need to been done by the authors, as follow:

1. In introduction part, the authors should illustrate the specific reason or application advantage of Dy3+ substitution on the structural, morphological and magnetic properties for ferrites. Meanwhile, the authors also should refer to the latest progress of the magnetoelectric nanocomposite. Therefore, it is quite necessary to add the related works or statements, and then elicit the research significance. To some extent, the cobalt-chromium ferrite is very conventional, and also lacks the novelty, so the authors should give the main reasons and consideration for the experimental design and explanation both in introduction and discussion parts.

2. According to this MS, the authors claim their mechanical and magnetoelectric properties is related to the structural characterization results, but it is possibly hard to conclude or understand the further evidence for their properties improvement. Hereby, I strongly suggest the author should build a correlation structure-property model for Dy3+ doped cobalt-chromium ferrite, and supplement the related statement to provide the directive evidence for the final explanation.

3. I have just listed a few but not all plotting problems and typing mistakes below:

Figure 1-8:

I suggest the authors should redraw and export all the figures with a high-resolution and more clear images.

Figure 9-12:

I strongly suggest the authors should reset the image scale and ratio, especially for Figure 9 and Figure 12.

In addition, the different variant must be italic all over the MS, such as “HDC”, “KT” and “K0”…, please have a check for all of them. And the formula (1) and (3) should be retype.

Conclusions part:

I strongly suggest that the author should supplement the 2~3 items of conclusion sentences for quick information and reading.

References:

The journal abbreviation, initial letter and typesetting in references should be done strictly according to the demand and rule of Nanomaterials. i.e. Ref.[1], Ref.[5], Ref.[17], Ref.[19], Ref.[22], Ref.[26], Ref.[28], Ref.[29], Ref.[40], Ref.[41], etc.

In addition, the author should add some related references for magnetoelectric nanocomposite too.

English writing:

English writing of this MS could be improved further.

……

Author Response

Comment #1: In introduction part, the authors should illustrate the specific reason or application advantage of Dy3+ substitution on the structural, morphological and magnetic properties for ferrites. Meanwhile, the authors also should refer to the latest progress of the magnetoelectric nanocomposite. Therefore, it is quite necessary to add the related works or statements, and then elicit the research significance. To some extent, the cobalt-chromium ferrite is very conventional, and also lacks the novelty, so the authors should give the main reasons and consideration for the experimental design and explanation both in introduction and discussion parts.

Reply: Indeed a good suggestion, we have now thoroughly revised the introduction part according to the comments.

Comment #2: According to this MS, the authors claim their mechanical and magnetoelectric properties is related to the structural characterization results, but it is possibly hard to conclude or understand the further evidence for their properties improvement. Hereby, I strongly suggest the author should build a correlation structure-property model for Dy3+ doped cobalt-chromium ferrite, and supplement the related statement to provide the directive evidence for the final explanation.

Reply: Improvement in the magnetization and coercivity characteristics is observed in CoCr0.5DyxFe1.5-xO4 ferrite system upon Dy3+ substitution. These improvements is discussed in relation to the magnetic moment, cation distribution and magnetic exchange interaction among the constituent ions in CoCr0.5DyxFe1.5-xO4 ferrite system. Directive evidence is provided by the M-H loops measurements carried out on vibrating sample magnetometer. Further, the appropriate sample (CoCr0.5Dy0.05Fe1.45O4) based on structural and magnetic properties was chosen as a ferromagnetic-magnetostrictive component in the mixed ME composite material. This sample further composed with BaTiO3 for its ME measurements. The ME measurements were carried out on ME setup (described in methods and materials part) for the directive evidence.

Comment #3: I have just listed a few but not all plotting problems and typing mistakes below:

Figure 1-8:

I suggest the authors should redraw and export all the figures with a high-resolution and more clear images.

Figure 9-12:

I strongly suggest the authors should reset the image scale and ratio, especially for Figure 9 and Figure 12.

In addition, the different variant must be italic all over the MS, such as “HDC”, “KT” and “K0”…, please have a check for all of them. And the formula (1) and (3) should be retype.

Reply: We have now improved the quality of figures. Different variant made italic all over the manuscript. Formula (1) and (3) is retyped.

Comment #4: Conclusions part:

I strongly suggest that the author should supplement the 2~3 items of conclusion sentences for quick information and reading.

Reply: Conclusion part is now revised according to the comment.

Comment #5: References:

The journal abbreviation, initial letter and typesetting in references should be done strictly according to the demand and rule of Nanomaterials. i.e. Ref.[1], Ref.[5], Ref.[17], Ref.[19], Ref.[22], Ref.[26], Ref.[28], Ref.[29], Ref.[40], Ref.[41], etc.

Reply: Reference section is now formatted according to the demand and rule of Nanomaterials.

Comment #6: In addition, the author should add some related references for magnetoelectric nanocomposite too.

Reply: References related to the magnetoelectric nanocomposite are now included in the manuscript.

Comment #7: English writing:

English writing of this MS could be improved further.

Reply: We have taken the assistance of native English speaker and improved the English writing of the manuscript.

Reviewer 5 Report

The number of possibilities of CoFe2O4 doping is infinite, thus there is no novelty in the paper and it seems to be a routine work which has a minimal impact. I recommend to reject this manuscript  for the following reasons.

1)     Most of the article is based on the XRD patterns shown in Fig 1. The experimental details show, that each pattern scanwas measured around 15 minute only (page 2 line 84). From such a fast scan, all parameters deduced (and the long discussion) are not accurate. The statistic is too low. A much longer time scan (at least 10-15 hours) is needed for accurate data

2)     No uncertainty-- values are provided neither in the text nor in Tables, thus all various  listed data (the lattice parameters, Hc Ms e.g.) may be within the uncertainties-the same.

3)     No saturations are achieved in M(H) plots Fig.10, thus the data in Table 6. (without uncertainties) are misleading.

Author Response

Comment #1: Most of the article is based on the XRD patterns shown in Fig 1. The experimental details show, that each pattern scanwas measured around 15 minute only (page 2 line 84). From such a fast scan, all parameters deduced (and the long discussion) are not accurate. The statistic is too low. A much longer time scan (at least 10-15 hours) is needed for accurate data

Reply: XRD was carried out at the rate of 1.5 °/min and almost took 40 min to complete the measurements. Since the XRD is seems to be clean we decided the scanning rate is sufficient for our sample. We agree with the reviewer that longer time scan is needed for accurate data, however we believe that the scanning rate is sufficient for our samples. In general, 10-15 hours of scanning for one sample is quite time consuming.

Comment #2: No uncertainty-- values are provided neither in the text nor in Tables, thus all various listed data (the lattice parameters, Hc Ms e.g.) may be within the uncertainties-the same.

Reply: We have now included the estimated error values in the table.

Comment #3: No saturations are achieved in M(H) plots Fig.10, thus the data in Table 6. (without uncertainties) are misleading.

Reply: We have now included the estimated error value in Table 6.

Round 2

Reviewer 2 Report

Fig. 12. I still think it is important to show the data of the ME coefficients with/without the Dy substitution in Co-Cr ferrite/BTO composites.

Author Response

Reply: We have now included the data of ME coefficients with/without the Dy substitiuton in Co-Cr ferrite/BTO composites in Fig. 12. Three compositions of ferrite CoCr0.5Fe1.5O4, CoCr0.5Dy0.05Fe1.45O4 and CoCr0.5Dy0.1Fe1.4O4 were chosen as ferromagnetic component phase for its nanocomposite with ferroelectric BaTiO3.

Reviewer 3 Report

All issues were well addressed, and this work can be accepted in the present form.

Author Response

Thank you so much for the comments. Your comments were very helpful to improve the quality of our manuscript.

Reviewer 4 Report

The authors have well answered my questions and comments in their revised MS, but I agree that this MS could be published on Nanomaterials after English improvement and polishing again.

Author Response

The authors have well answered my questions and comments in their revised MS, but I agree that this MS could be published on Nanomaterials after English improvement and polishing again.

Reply: We sincerely appreciate the reviewer's comments. We have now again taken the assistance of two native English speakers that helped us to carefully revised and improved the English writing of the manuscript. We hope that the level of English language of our manuscript is now fairly close to the standards set by MDPI.

Reviewer 5 Report

Thye authors did not reply to two of the comments

1) A better XRD patterns are needed to deduce the exact crystallographic data

3) The authors use in Table 6 the saturation magnetization (see line 374) whereas no saturation is achieved in the Figure

Author Response

Comment: A better XRD patterns are needed to deduce the exact crystallographic data.

Reply: We respect the comments of reviewer. XRD was carried out at the rate of 1.5 °/min and almost took 40 min to complete the measurements. Since the XRD is seems to be clean we decided the scanning rate is sufficient for our sample. We agree with the reviewer that longer time scan is needed for accurate data, however we believe that the scanning rate is sufficient for our samples. In general, 10-15 hours of scanning for one sample is quite time consuming.

We are apologize and regret that we could not do the XRD measurements again. We are requesting to the reviewer to understand the situation and favour us on this issue.   

Comment: The authors use in Table 6 the saturation magnetization (see line 374) whereas no saturation is achieved in the Figure.

Reply: Indeed a good observation, accordingly we have now revised the manuscript. A maximum magnetic field of ±15 kOe was applied for the M-H curve measurements. Magnetization not completely saturated even at ±15 kOe. Table 6 lists the magnetization at 15 kOe (Ms), remanent magnetization (R), and coercivity (Hc) obtained from the magnetization curves.